# Fire Safety Knowledge of Firefighting Equipment among Local and Foreign University Students

**DOI:** 10.3390/ijerph191912239

**Published:** 2022-09-27

**Authors:** Chu Zhang, Won-Hwa Hong, Young-Hoon Bae

**Affiliations:** 1School of Architectural, Civil, Environmental and Energy Engineering, Kyungpook National University, Daegu 41566, Korea; 2Research Institute of Intelligent Fire Safety Technology and Human Behavioral Science, Pukyong National University, Busan 48513, Korea

**Keywords:** fire safety knowledge, university students, firefighting equipment

## Abstract

Despite the risks at university laboratories, university students are still marginalized from safety management in university laboratories. In addition, the existing studies related to the fire safety knowledge of university laboratories, do not consider the fire safety knowledge of university students with respect to firefighting equipment and the increasing number of foreign university students. To overcome this gap, we conducted a survey on 273 foreign and 144 local students and identified the differences in fire safety knowledge and those in comprehension and response related to firefighting equipment among the participants through statistical analysis. The results of the survey, where respondents were classified into four groups by gender and nationality, found significant differences in fire safety knowledge between gender and nationality. All the groups had difficulty in directly extinguishing a fire using fire extinguishing equipment. The results of this study, that is, those pertaining to the differences in fire safety knowledge depending on the gender and nationality of students and types of firefighting systems are expected to be used as basic data to establish safety education and management plans in the future.

## 1. Introduction

Academic laboratories in universities likely face various risks from chemical, biological, physical, and artificial factors [1,2,3,4]. Previous studies indicate that laboratories in universities are more dangerous than industrial laboratories because the former do not implement rigid safety management practices, promote a safety culture, or invest adequately in safety [5,6].

The propensity of university laboratories to such risks is confirmed by various statistical data and accident cases as well. According to the ‘2019 Report on the Status of Safety Management in Laboratories’, 62% of all the laboratory accidents reported in the Republic of Korea in 2018 (87 out of 140 accidents) occurred in university laboratories [7]. In addition, according to U.S. fire statistics, 1726 fires occurred in university campuses and were associated with 43 injuries or deaths [8]. Further, in China, one postdoctoral researcher died following an explosion that occurred in a chemical laboratory in 2015 [9] and three students died in a laboratory explosion in 2018 [10]. Similar laboratory accidents resulting in casualties continue to occur worldwide.

Despite the risk of such university laboratories and the fact that most university laboratories are located in the building where university students’ lectures are, university students are still excluded from the safety management of university laboratories. 

In addition, very few studies have been conducted on the safety knowledge of university students. To date, some studies have analysed the safety knowledge of universities and university laboratories. Gong [11] analysed the impact of students’ major, grade, and gender on institutions’ safety culture by surveying 362 university students in Beijing, China, and proposed measures to improve safety. Further, Hasan and Younos [12] surveyed 964 university students in Bangladesh and analysed the differences in safety culture according to gender and residential area (resident and non-residents students). Similarly, Armstrong [13] investigated the safety culture of graduate students in a university’s chemistry department by conducting a survey. This study analysed the graduate students’ safety attitudes and safety behaviours and proposed measures to improve the institution’s safety culture. Kong et al. [14] conducted a safety culture survey on 231 postgraduate nursing students from medical schools in China. They revealed that the students had low positive attitudes towards patient safety culture and the students’ safety attitudes were affected by age, work experience, workload, and safety education. 

Although they analysed differences in the safety knowledge and cultures of undergraduate and graduate students using various variables (e.g., major, grade, and gender), these studies have the following limitations: First, no study considered the fire safety knowledge of university students with respect to firefighting equipment, which are directly associated with response in the event of an accident. In other words, previous studies focused on the safety culture of students’ real life (e.g., ‘I often consider various safety problems within the campus’ and ‘I never used forbidden high-powered electric equipment inside my dormitory’), and none of them analysed the fire safety knowledge related to firefighting equipment, which is directly related to students’ response to an accident. Second, they did not consider the impact of an increase in the number of foreign students, which is being observed worldwide, on safety culture [15,16,17,18]. In general, foreign students exhibit cultural differences compared to local students [19,20,21,22] and have lower language skills [23,24]. Therefore, fire safety knowledge is expected to differ among foreign and local students. Hence, in an accident situation where students’ rapid initial response is paramount, the difference in fire safety knowledge between foreign and local students is highly likely to have a negative impact on the initial response. However, to date, no study has considered the difference in fire safety knowledge between foreign and local students.

To overcome the limitations of previous studies, this study analysed the fire safety knowledge (comprehension and response) of local and foreign university students with respect to firefighting equipment. To this end, a survey was conducted on 417 students (273 foreign and 144 local students). Subsequently, the differences in fire safety knowledge related to firefighting equipment among the participants were presented through statistical analysis. This study is significant in that it analysed the fire safety knowledge of both local and foreign university students. The results of this study can be used as basic data to establish measures to improve the university students’ fire safety knowledge.

## 2. Materials and Methods

### 2.1. Survey Target and Design

The fire safety knowledge for firefighting equipment was analysed by dividing it into comprehension and safety response for firefighting equipment. Here, comprehension means to ‘know what the firefighting equipment are’ and ‘know how to use the firefighting equipment’, and response means to have the knowledge of ‘whether firefighting facilities can be used in a fire’. The composition of the questionnaire is shown in Table 1. The survey included 35 questions, among which five were related to personal information, 20 to comprehension for firefighting equipment, and 10 to response for firefighting equipment (see Table 1). The questionnaire on fire safety knowledge comprised 35 questions on comprehension and response for the 10 firefighting equipment (two fire extinguishing equipment, and eight escape equipment) that must be installed in universities under Republic of Korea law [25,26]. At this time, among the 14 items of firefighting equipment that must be installed legally in Korea, 4 pieces of equipment not used by university students (emergency alert equipment, outdoor fire hydrant, automatic fire detection equipment, and emergency broadcasting equipment) were excluded. Further, 20 questions (Q6) related to comprehension were composed of two questions (‘I know what the (name of firefighting equipment) is’ and ‘I know how to use (name of firefighting equipment)’) to examine whether the students were aware of the appearance of the 10 items of firefighting equipment and knew how to use them. In addition, the 10 questions (Q7) related to response are composed of one question (‘I can use (name of firefighting equipment) in the event of the fire’) to examine whether the students can utilise 10 items of firefighting equipment in a disaster situation.

The responses to the questionnaire for fire safety knowledge analysis was assessed using a five-point Likert scale (with values ranging from 1 = strongly disagree to 5 = strongly agree). In addition, comprehension and response scores were presented as average scores of the relevant questions, and the fire safety knowledge score was calculated as the sum of the comprehension and response scores (fire safety knowledge scores = comprehension scores + response scores). The survey was conducted on 417 randomly selected students (273 foreign and 144 local students) at Kyungpook National University in the Republic of Korea. Safety culture (including fire safety knowledge) is influenced by the respondent’s background (e.g., country) [27,28,29,30,31], meaning that in this survey, it is not possible to assume all foreign students are of the same group. Therefore, in this study, the scope of foreign students was limited to Chinese students, who account for 43.6% of foreign students in Korea [32], and survey respondents were recruited accordingly. They were recruited through recruitment advertising on social network services (SNS) and online bulletin boards. The survey was conducted online (office.naver.com).

Also, the purpose of this paper is to analyze the safety knowledge of university students who are currently neglected in laboratory safety management. In this paper, the survey respondents were limited to university students and a survey was conducted. All the respondents participated in the survey after recognising the study’s goal and providing written consent to participate in the study. Table 2 summarises the demographic information of all the survey respondents.

The survey respondents were 273 foreign students (65.5%) and 144 local students (34.5%). They included 222 male (53.2%) and 195 female (46.8%) students. For questions related to language skills and years of residence for foreign students, 159 foreign students (58.2%) reported high language skills (advanced), and the highest proportion of foreign students (34.1%) reported four to six years of residence (96 students). Furthermore, it was reported that 69.1% of respondents completed two or four hours of safety education.

### 2.2. Data Analysis

To clarify the fire safety knowledge, comprehension, and response of local and foreign university students regarding firefighting equipment, the results of the survey were analysed in three steps:

Internal consistency analysis (calculation of Cronbach’s alpha) was conducted to evaluate the questionnaire’s reliability. The analysis was conducted according to the items of the questionnaire, for each of the following: fire safety knowledge, comprehension, and response for firefighting equipment.

Statistical analysis was conducted on the differences in fire safety knowledge depending on the survey respondents’ personal information (gender, nationality, years of residence, language skills, and hours completed in safety education). Accordingly, the *t*-test was conducted on gender and nationality, which had two variables each, and analysis of variance (ANOVA) was conducted on years of residence, language skills, and hours completed in safety education, each of which had three or more variables. All the statistical analyses were conducted on fire safety knowledge, comprehension, and response for firefighting equipment. After the statistical analysis, respondents were divided into groups according to the aspects of their personal information that affected fire safety knowledge.

Statistical analysis (the *t*-test) was conducted to analyse the difference between comprehension and response for each respondent group. Finally, a statistical analysis (the *t*-test) was conducted on the difference between comprehension and response depending on the type of firefighting equipment (fire extinguishing equipment, alarming equipment, and escape equipment) for each respondent group. In this study, the statistical processing and analysis of all survey data were conducted using IBM SPSS Statistics 25 software.

## 3. Results

### 3.1. Difference in Fire Safety Knowledge among Firefighting Equipment depending on Respondents’ Demographic Information

Prior to conducting statistical analysis, an internal consistency analysis was conducted to evaluate the reliability of the survey. Table 3 presents the results of this consistency analysis. As shown in the table, the value of Cronbach’s alpha exceeded 0.9 (comprehension:0.947, response: 0.904, and fire safety knowledge: 0.964), which indicates that the survey’s reliability and consistency were acceptable.

Subsequently, statistical analysis was conducted on the differences in fire safety knowledge depending on the respondents’ personal information (gender, nationality, years of residence, and language skills). The differences in fire safety knowledge depending on gender and nationality, which had two variables, were analysed using the *t*-test (see Table 4), whereas those in fire safety knowledge depending on years of residence, language skills, and hours completed in safety education, which had three variables, were analysed using ANOVA (see Table 5). The *t*-test results revealed that all the comprehension, response, and fire safety knowledge scores for firefighting equipment exhibited significant differences depending on gender (*p* < 0.05). Further, the scores of male respondents were higher than those of female respondents (see Table 4).

Regarding the comprehension and fire safety knowledge scores for firefighting equipment according to nationality, foreign students exhibited higher scores than local students; however, no statistically significant difference was observed among the scores (*p* > 0.05). 

Regarding the comprehension score of foreign students according to years of residence, ‘more than 6 years’ exhibited the highest score of 3.375; it was followed by ‘2–4 years’, ‘4–6 years’, and ‘less than 2 years’. For response score according to years of residence, ‘more than 6 years’ indicated the highest score; it was followed by ‘2–4 years’, ‘4–6 years’, and ‘less than 2 years’. In other words, an increase in comprehension, response, and fire safety knowledge scores was observed with an increase in years of residence; however, there was no significant difference among scores based on ANOVA results (*p* > 0.05) (see Table 5).

For all the scores of fire safety knowledge, comprehension, and response according to the language skills of foreign students, ‘intermediate’ exhibited the highest scores. However, there was no significant difference in comprehension, response, and fire safety knowledge scores based on ANOVA results (*p* > 0.05) (see Table 5). In addition, for all the scores of fire safety knowledge, comprehension, and response according to the hours completed in safety education, ‘6 h’ exhibited the highest scores. However, there was no significant difference in comprehension, response, and fire safety knowledge scores based on ANOVA results (*p* > 0.05) (see Table 5).

Therefore, the statistical analysis results revealed significant differences in fire safety knowledge score, depending on the gender aspects of the respondents’ personal information, but no significant difference depending on the respondents’ nationality, years of residence, language skills and hours completed in safety education was noticed.

Additionally, in order to clarify the difference between local and foreign university students, which is the purpose of this paper, respondents were divided into four groups based on gender which were found to have the greatest influence on fire safety knowledge and respondents’ nationality, and statistical analysis was conducted on the differences in comprehension, response, and fire safety knowledge scores for firefighting equipment across the groups, as shown in Table 6. In this case, each group was named using the first letter of the group nationality (whether local or foreign) and the first letter of the gender (male or female).

Group F-M exhibited the highest fire safety knowledge scores for firefighting equipment, followed by groups L-M, L-F, and F-F. ANOVA results revealed significant differences among the four groups (*p* < 0.05) (Table 6).

### 3.2. Difference between Comprehension and Response for Firefighting Equipment

A statistical analysis (*t*-test) was conducted to analyse the difference between comprehension and response among the fire safety knowledge for firefighting equipment (see Table 7).

For all survey respondents, the average score for comprehension (3.186) was higher than that of response (3.040). In addition, for fire extinguishing equipment, the average score for comprehension (3.529) was higher than that of response (3.155), and there was a significant difference among scores based on the statistical verification results (*p* < 0.05). Finally, for the escape equipment, the average score for comprehension (3.100) was higher than that of response (3.012); however, there was no significant difference among scores based on the statistical verification results (*p* > 0.05).

Further, for group L-M and F-M, the average score for comprehension was higher than that of response. Similarly, for fire extinguishing equipment and escape equipment, the average scores of the comprehension were higher than those of response; however, there was no significant difference among scores based on the statistical verification results (*p* > 0.05). For group L-F, F-F, the average score for comprehension was higher than that of response. Similarly, for fire extinguishing equipment and escape equipment, the average scores of the comprehension were higher than those of response. Among them, there was a significant difference among scores by the average scores of fire extinguishing equipment based on the statistical verification results (*p* < 0.05). 

In order words, for all the four groups, the comprehension score was higher than the response score for fire extinguishing equipment and escape equipment. Among them, only the fire extinguishing equipment of group L-F and F-F showed a statistically significant difference.

## 4. Discussion

In recent years, following the occurrence of several fires in university laboratories, various studies were conducted to promote the safety culture among students [13,14]. However, no study has yet been conducted on foreign students or considered the fire safety knowledge of firefighting equipment. 

Therefore, in this study, a survey was conducted to investigate the fire safety knowledge (comprehension and response) of foreign and local students for firefighting equipment. It revealed that the comprehension and response scores of foreign and local students for firefighting equipment were moderate (according to the mean scores of each index), which indicated that the fire safety knowledge of foreign and local university students in the Republic of Korea was not adequately practiced and promoted. These results are similar to those reported by some earlier studies [11,12]. The survey results revealed that the fire safety knowledge score of male respondents (6.547) was higher than that of female respondents (5.862) (see Table 4). This finding is different from the results of earlier studies: According to Hasan and Younos [12], the Safety culture score of female respondents (3.72) was higher than that of male respondents (3.58). Similarly, Gong [11] revealed that the Safety culture score of female respondents (3.812) was higher than that of male respondents (3.716). It appears that the results of this study, which were obtained through a survey, contrasted those of earlier studies because the survey analysed the fire safety knowledge (comprehension and response) for firefighting equipment. Earlier studies prepared questionnaires with a focus on the safety culture of real life, for instance, ‘I often consider various safety problems within the campus’ and ‘I never used forbidden high-powered electric equipment in my dormitory’ [11,12]. However, in the current study, the survey questionnaire included questions related to the recognition (comprehension) and use (response) of firefighting equipment, such as ‘I know what the (firefighting equipment) are’ and ‘I can use (firefighting equipment) in the event of an accident’. It appears that this difference caused the fire safety knowledge score of male respondents to be higher than that of female respondents in this study. The difference in fire safety knowledge for firefighting equipment depending on gender, which was observed in this study, indicates the necessity of establishing different measures for each gender at the time of implementing safety education and safety policies (see Table 4).

The study also revealed that the fire safety knowledge score of foreign students was higher than that of local students. In the survey planning stage, it was expected that the cultural differences exhibited by foreign students and their lower language skills compared to local students would have a negative impact on the fire safety knowledge score; however, the survey results indicate the opposite trend. Further, this result contradicts an earlier study which found that foreigners have lower fire safety knowledge compared to local people [33]. This indicates that there are factors that have a positive influence on the fire safety knowledge of foreign students, unlike foreign workers, and necessitates further research on this topic (see Table 4). The difference in safety education and safety culture in the home country of foreign students (in this paper, Chinese students) can be considered as the cause of this result. However, there are no studies comparing the safety culture of middle and high school students in China and Korea. In addition, the safety education regulations for middle and high school students in Korea and China both stipulate the same twice a year. In the future, it is judged that a study comparing the safety education methods of China and Korea is necessary to verify their effectiveness [34,35].

In addition, the survey results showed that language skills, years of residence, and hours completed in safety education did not have a significant influence on fire safety knowledge. The finding that years of residence do not have a significant influence on fire safety knowledge is similar to the result of an earlier study [11,12] (see Table 5).

Lastly, the safety knowledge of university students showed a trend of increasing as the number of hours completed in safety education increased but showed the lowest result when it exceeded 6 h. This is a very interesting result considering that fire safety knowledge showed a trend to increase according to the number of years of residence. This suggests the need for additional analysis of fire safety knowledge according to the method and number of safety training. (see Table 5).

Finally, there was a significant difference between the comprehension and response scores for firefighting equipment. When the scores of the four groups, which were classified based on gender and nationality, were compared, the comprehension score exceeded 3 points for all the groups, whereas the response score was lower than 3 points for group L-F and F-F. In addition, female respondents exhibited a larger difference between comprehension and response scores compared to their male counterparts. This means that female respondents experienced greater difficulty in using firefighting equipment.

The result that the comprehension and response scores for fire extinguishing equipment was higher than those for escape equipment in all groups means that the respondents’ fire safety knowledge of fire extinguishing equipment was generally higher (see Table 7). In addition, the result that the statistical difference between the comprehension and response of female groups appeared only in the fire extinguishing equipment means that the female groups experienced difficulties in directly extinguishing fire using the fire extinguishing equipment. It is necessary to establish the direction of safety education based on these findings.

Based on these results, it is necessary to establish the direction of safety education, for example, education to increase understanding of escape equipment for all groups, or education to teach women in the use of fire extinguishing equipment.

## 5. Conclusions

In this study, the fire safety knowledge of local and foreign university students related to firefighting equipment was analysed using a survey. The results provide empirical evidence for the fire safety knowledge of foreign and local university students in the Republic of Korea. The main results of this study are summarised as follows.

The fire safety knowledge scores of foreign and local university students were moderate.Male students exhibited a higher fire safety knowledge score with respect to firefighting equipment compared to female students.Foreign students showed a higher fire safety knowledge score with respect to firefighting equipment compared to local university students.For all the groups, comprehension scores were higher than response scores for firefighting equipment.Female students had difficulty in directly extinguishing the fire using fire extinguishing equipment.

This study is significant as, unlike previous studies, it analysed and provided comprehension and response scores for firefighting equipment. The results of this study pertaining to the differences in fire safety knowledge depending on the gender and nationality of students and types of firefighting equipment could be used as basic data to establish safety education and management plans in the future.

## Figures and Tables

**Table 1 ijerph-19-12239-t001:** Questionnaire design.

Questionnaire Item			Question	
Personal information			Q 1	Nationality
			Q 2	Gender
			Q 3	Language skill
			Q 4	Years of residence
			Q 5	Hours completed in safety education
Fire Safety Knowledge	C	F	Q 6–1	I know what the ‘*fire extinguisher*’ is.
			Q 6–2	I know how to use ‘*fire extinguisher*’.
			Q 6–3	I know what the ‘*indoor fire hydrant*’ is.
			Q 6–4	I know how to use ‘*indoor fire hydrant*’.
		E	Q 6–5	I know what the ‘*escape ladder*’ is.
			Q 6–6	I know how to use ‘*escape ladder*’.
			Q 6–7	I know what the ‘*rescue spiral chute*’ is.
			Q 6–8	I know how to use ‘*rescue spiral chute*’.
			Q 6–9	I know what the ‘*descending lifeline*’ is.
			Q 6–10	I know how to use ‘*descending lifeline*’.
			Q 6–11	I know what the ‘*escape guidance line*’ is.
			Q 6–12	I know how to use ‘*escape guidance line*’.
			Q 6–13	I know what the ‘*exit signs*’ is.
			Q 6–14	I know how to use ‘*exit signs*’.
			Q 6–15	I know what the ‘*escape route signs (corridor)*’ is.
			Q 6–16	I know how to use ‘*escape route signs (corridor)*’.
			Q 6–17	I know what the ‘*escape route signs (seat)*’ is.
			Q 6–18	I know how to use ‘*escape route signs (seat)*’.
			Q 6–19	I know what the ‘*phosphorescent guidance line*’ is.
			Q 6–20	I know how to use ‘*phosphorescent guidance line*’.
	R	F	Q 7–1	I can use ‘*fire extinguisher*’ in the fire.
			Q 7–2	I can use ‘*indoor fire hydrant*’ in the fire.
		E	Q 7–3	I can use ‘*escape ladder*’ in the fire.
			Q 7–4	I can use ‘*rescue spiral chute*’ in the fire.
			Q 7–5	I can use ‘*descending lifeline*’ in the fire.
			Q 7–6	I can use ‘*escape guidance line*’ in the fire.
			Q 7–7	I can use ‘*exit signs*’ in the fire.
			Q 7–8	I can use ‘*escape route signs (corridor)*’ in the fire.
			Q 7–9	I can use ‘*escape route signs (seat)*’ in the fire.
			Q 7–10	I can use ‘*phosphorescent guidance line*’ in the fire.

C: comprehension; R: response; F: fire extinguishing equipment; E: escape equipment; The questionnaire comprised two questions about comprehension and one question about response for 10 items of firefighting equipment that should be installed in all universities under Republic of Korea law. The 10 items of firefighting equipment are as follows: fire extinguishing equipment (fire extinguisher, indoor fire hydrant); and escape equipment (escape ladder, rescue spiral chute, descending lifeline, escape guidance line, exit signs, escape route signs (corridor), escape route signs (seat), and phosphorescent guidance line); The Participants’ language skills were classified based on their scores in the authorised Republic of Korean proficiency test (levels 1 and 2: beginner, levels 3 and 4: intermediate, and levels 5 and 6: advanced). Furthermore, years of residence were classified into ‘less than 2 years’, ‘2–4 years’, ‘4–6 years’, and ‘more than 6 years’ based on the number of years foreign students had lived in Republic of Korea. And Hours completed in safety education were classified into ‘2 h’, ‘4 h’, ‘6 h’, and ‘more than 6 h’.

**Table 2 ijerph-19-12239-t002:** Demographic information of survey respondents.

Category		Percentage (Number) of		
		All Students (N = 417)	Local Students (N = 144)	Foreign Students (N = 273)
Gender	Male	53.2 (222)	60.4 (87)	49.5 (135)
	Female	46.8 (195)	39.6 (57)	50.5 (138)
Language skill	Beginner			5.5 (15)
	Intermediate			36.3 (99)
	Advanced			58.2 (159)
Years of residence	Less than 2 years			12.1 (33)
	2–4 Years			19.8 (54)
	4–6 Years			34.1 (96)
	More than 6 years			33.0 (90)
Hours completed in safety education	2 h	36.0 (150)	31.2 (45)	38.4 (105)
	4 h	33.1 (138)	43.8 (63)	27.5 (75)
	6 h	10.0 (42)	6.2 (9)	12.1 (33)
	More than 6 h	20.9 (87)	18.8 (27)	22.0 (60)

**Table 3 ijerph-19-12239-t003:** Analysis results of the internal consistency of the survey questionnaire.

Category	M	S.D	Item (n)	IIC Score
Comprehension	3.186	0.773	20	0.947
Response	3.040	0.789	10	0.904
Fire Safety Knowledge	6.226	1.542	30	0.964

M: mean; S.D: standard deviation; IIC: Cronbach’s alpha.

**Table 4 ijerph-19-12239-t004:** Differences in fire safety knowledge, comprehension, and response score and *t*-test results according to respondents’ gender and nationality.

Category		Fire Safety Knowledge				Comprehension				Response			
		M	S.D	t	*p*	M	S.D	t	*p*	M	S.D	t	*p*
G	Male	6.547	1.525	2.671	0.008 *	3.337	0.754	2.507	0.013 **	3.209	0.784	2.761	0.007 *
	Female	5.862	1.490			3.014	0.765						
N	Local	6.068	1.524	0.885	0.381	3.151	0.782	0.386	0.700	2.917	0.763	1.346	0.181
	Foreign	6.310	1.553			3.204	0.772			3.105	0.799		

G: Gender; N: Nationality; M: mean; S.D: standard deviation; t: t-value; *p*: *p*-value; * *p* < 0.01; ** *p* < 0.05.

**Table 5 ijerph-19-12239-t005:** Differences in fire safety knowledge, comprehension, and response scores and ANOVA results according to years of residence and language skill.

Category		Fire Safety Knowledge				Comprehension				Response			
		M	S.D	F	*p*	M	S.D	F	*p*	M	S.D	F	*p*
Years of residence	Less than 2 years	5.568	1.517	1.252	0.296	2.768	0.777	1.735	0.166	2.800	0.755	0.825	0.483
	2–4 Years	6.356	1.703			3.244	0.862			3.111	0.859		
	4–6 Years	6.253	1.745			3.172	0.831			3.081	0.929		
	More than 6 years	6.615	1.196			3.375	0.603			3.240	0.611		
Language skill	Beginner	6.330	1.402	0.514	0.600	3.170	0.740	0.647	0.526	3.160	0.666	0.396	0.674
	Intermediate	6.092	1.745			3.086	0.851			3.006	0.912		
	Advanced	6.443	1.450			3.281	0.727			3.162	0.740		
Hours completed in safety education	2 h	6.241	1.406	0.459	0.712	3.183	0.702	0.537	0.658	3.058	0.729	0.380	0.768
	4 h	6.336	1.599			3.247	0.798			3.089	0.821		
	6 h	6.400	1.220			3.300	0.614			3.100	0.636		
	More than 6 h	5.943	1.827			3.040	0.922			2.903	0.917		

M: mean; S.D: standard deviation; F: f-value; *p*: *p*-value.

**Table 6 ijerph-19-12239-t006:** Fire safety knowledge, comprehension, and response scores and the ANOVA results of each respondent group.

Case	Fire Safety Knowledge				Comprehension				Response			
	M	S.D	F	*p*	M	S.D	F	*p*	M	S.D	F	*p*
Group L-M	6.740	1.442	3.023	0.032 **	3.416	0.713	2.513	0.061	3.324	0.740	3.590	0.015 **
Group L-F	5.889	1.557			2.998	0.779			2.891	0.804		
Group F-M	6.247	1.624			3.216	0.809			3.031	0.831		
Group F-F	5.795	1.351			3.053	0.749			2.742	0.627		

Group L-M: local male group; Group L-F: local female group; Group F-M: foreign male group; Group F-F: foreign female group; M: mean; S.D: standard deviation; F: f-value; *p*: *p*-value; ** *p* < 0.05.

**Table 7 ijerph-19-12239-t007:** Difference between Comprehension and Response scores depending on the types of firefighting equipment by group (*t*-test results).

Case	Firefighting Equipment	Comprehension		Response		t	*p*
		M	S.D	M	S.D		
Total	Total	3.186	0.773	3.040	0.789	1.555	0.121
	Fire extinguishing	3.529	0.841	3.155	0.959	3.459	0.001 *
	Escape	3.100	0.841	3.012	0.849	0.876	0.382
Group L-M	Total	3.416	0.713	3.324	0.740	0.595	0.554
	Fire extinguishing	3.789	0.805	3.467	0.894	1.797	0.076
	Escape	3.322	0.787	3.289	0.809	0.200	0.842
Group L-F	Total	2.998	0.779	2.891	0.804	0.646	0.520
	Fire extinguishing	3.293	0.803	2.859	0.953	2.367	0.020 **
	Escape	2.924	0.831	2.899	0.835	0.142	0.887
Group F-M	Total	3.216	0.809	3.031	0.831	0.857	0.395
	Fire extinguishing	3.655	0.828	3.431	0.832	1.029	0.308
	Escape	3.106	0.875	2.931	0.916	0.742	0.461
Group F-F	Total	3.053	0.749	2.742	0.627	1.386	0.174
	Fire extinguishing	3.289	0.871	2.711	0.976	0.821	0.012 **
	Escape	2.994	0.879	2.750	0.758	0.915	0.366

M: mean; S.D: standard deviation; Group L-M: local male group; Group L-F: local female group; Group F-M: foreign male group; Group F-F: foreign female group; t: t-value; *p*: *p*-value; * *p* < 0.01; ** *p* < 0.05.

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
