# Peer review of "Fire Safety Knowledge of Firefighting Equipment among Local and Foreign University Students"

_ijerph, 2022, doi:10.3390/ijerph191912239_

Round 1
Reviewer 1 Report (Previous Reviewer 2)
The authors have made some improvements. I don't have any more questions.
Reviewer 2 Report (Previous Reviewer 3)
I accept the corrections presented in accordance with the recommendations of the reviewers. However, I believe that removing references to the term "safety culture" from the publication has changed its meaning. Furthermore, in my opinion "safety culture" was an important aspect of the manusccript, and the proposed change to "safety management" is acceptable but limits the subject of the publication.
This manuscript is a resubmission of an earlier submission. The following is a list of the peer review reports and author responses from that submission.
Round 1
Reviewer 1 Report
Reference numbers in the body of the article must be modified according to the MDPI Paperwork Rules.
There are the following problems with respect to the scope of safety culture reviewed for the purpose of minimizing human casualties in the event of a fire.
- The remarks of SA and SB classified in Table 1 should be specifically described in accordance with the gist of the questionnaire.
- SA and SB of safety culture should be limited to firefighting equipment that can be actively used by evacuees (or residents). (Since automatic fire detection and emergency broadcasting equipment are manual equipment, it is reasonable to exclude them from the question.)
- Since the variables of this study vary greatly depending on the level of safety education conducted by universities, it is necessary to mention the basic level of safety education for foreign students.
- Since basic safety education in universities is divided by grade, the classification and interpretation of the subject's grade by grade should be included in the results.
- Since laboratory safety culture for laboratory safety is a separate field from university safety culture, a conclusion should be drawn through a separate analysis.
- As the interpretation of this study focuses on international students, the level of safety culture in the home country will act as a variable. Therefore, it needs to be supplemented.
- The question in Table 2 should reflect the number of hours completed in safety education at the university you are attending.
- The spacing between categories in Table 4 should be set uniformly for the same letter. Also, follow the rules for creating tables.
- Table 4 should be described at the bottom of the table for t and p. (Make sure no skills are missing.)
It is expected that the analysis will be clearer by reinforcing the above content in more detail.
Author Response
Please see the attachment. Many thanks.

Reviewer 2 Report
This is a meaningful study. However, this paper has a serious flaw: compared with the number of questions, the number of respondents is not enough. What's more, they are all from the same university. Therefore, the conclusion of this paper is one-sided and does not have general scientific significance. I suggest the author expand the scope of the survey, improve the results and conclusions and submit again.
Author Response
Please see the attachment. Many thanks.

Reviewer 3 Report
The manuscript describes research on safety culture elements such as safety attitude and safety behaviour of foreign and local groups of university students in relation to firefighting equipment.
The publication clearly states the purpose of the research. The scope is clearly described. The research group of 203 students from China and Republic of Korea, place-selected university and an extensive methodology, as well as the analysis of the results and their interpretation is properly presented. The manuscript shows how the research differs from the research conducted previously under similar conditions. The results obtained as a result of the survey research and their discussion in connection with the results published by other authors are clearly presented. The significant differences in the obtained results were noted (such as, the difference in safety culture for firefighting equipment depending on gender: male respondents score was higher than female), which was justified by the description of the issues concerning the details of the research topic.
The manuscript is written correctly. The division into chapters and their content is correct. The way of presenting the results is careful and clear. The selection of literature is sufficient. The language used is correct. The publication is completed with correct conclusions.
Only minor editing errors were noticed in the publication.
Line 182, Table 4. It should be checked if capital letter P is properly used in the middle of the second line on the table.
Line 202, Table 6 as above.
Author Response
Please see the attachments. Many thanks.
